# An On-Demand Dissoluble Chitosan Hydrogel Containing Dynamic Diselenide Bond

**DOI:** 10.3390/gels7010021

**Published:** 2021-02-20

**Authors:** Xingxia Xu, Weihong Lu, Jian Zhu, Xiangqiang Pan, Xiulin Zhu

**Affiliations:** 1State and Local Joint Engineering Laboratory for Novel Functional Polymeric Materials, Jiangsu Key Laboratory of Advanced Functional Polymer Design and Application, Department of Polymer Science and Engineering, College of Chemistry, Chemical Engineering and Materials Science, Soochow University, Suzhou 215123, China; xingxiaxu@163.com (X.X.); xlzhu@suda.edu.cn (X.Z.); 2State Key Laboratory of Radiation Medicine and Protection, School for Radiological and Interdisciplinary Sciences (RAD-X), Soochow University, Suzhou 215123, China

**Keywords:** chitosan, hydrogel, diselenide, on-demand dissolution

## Abstract

A new kind of on-demand dissolution hydrogel is successfully synthesized by modification of chitosan using γ-selenobutyrolactone. The chitosan hydrogel with different selenium contents is formed by ring opening of γ-selenobutyrolactone with the amines of D-glucosamine units on the chitosan backbone. The structure of the hydrogel was confirmed by ^1^H NMR, XRD and XPS. Its physical and biological properties were evaluated by rheology characterization, degradation tests and cytotoxicity test. The hydrogel showed excellent biocompatibility and good degradation properties under oxidation or reduction conditions. All the evidence demonstrated that this type of material has good prospects for dressing applications.

## 1. Introduction

Chitosan is a linear polysaccharide consists of randomly distributed β-(1→4)-linked D-glucosamine (deacetylated unit) and N-acetyl-D-glucosamine (acetylated unit). It is a derivative of chitin (poly-N-acetylglucosamine), which is the second most abundant biopolymer after cellulose [1]. Chitosan is a type of green and renewable material which was first found in 1811 by H. Braconnot from mushrooms [2]. Its related materials were found applications in wound dressing [3,4], tissue engineering scaffolds [5,6], biological suture [7], and drug delivery vehicles [8,9]. Since chitosan contains a large amount of amino and hydroxyl groups, it can form physical or chemical hydrogels via noncovalent and covalent bonds [10]. Physical crosslinks mainly include ionic/electrostatic interaction, hydrogen bonding, hydrophobic interaction, metal coordination, or host–guest interactions. Physically crosslinked chitosan hydrogels are stimuli-responsible with self-healing [11] and injectable properties [12] under room temperature, while chemically crosslinked chitosan hydrogels are normally mechanically stable and non-degradable. Up to now, the multi stimuli-responsive chitosan-hydrogels with better strength has received much attention. Some dynamic bonds, such as imine [13] and disulfide bonds [14,15], have been introduced into the chitosan-hydrogels. Although many researchers paid much attention to the degradation of hydrogels [16,17,18], _ENREF_17_ENREF_18seldom is it reported about the chitosan hydrogels materials with the property of dissolution on-demand [19]._ENREF_14 In the treatment of moderate and severe burns, the process of dressing change lead to traumatization of newly epithelialized tissues, delayed healing, and personal suffering in the injured patient. If hydrogel dressings could dissolve on demand, it would greatly reduce the pain of patients in the process of dressing change or debridement, and furthermore reduce the secondary injury of patients [20,21].

Se-containing polymers have redox properties owing to special electronegativity and atomic radius of selenium. Selenium is a biologically essential element for the human body and is incorporated in proteins to make selenoproteins which prevent cellular damage from free radicals [22,23]. _ENREF_24Diselenide compounds have been found many applications in antibacterial, chronic inflammation and prevention of cancer [24,25,26,27]. Moreover, diselenide bonds can be broken by oxidants or reductants, which have been in-depth studied in drug release systems [28,29,30,31]. _ENREF_26Our group has concentrated on design and synthesis of novel diselenide-containing polymers for a long time [32,33,34]._ENREF_29 Se-containing polymers with multiple topological structures have been prepared in our lab. More recently, we reported a diselenide-cross-linked nanofiber based on poly(2-oxazoline), which can dissolve under UV irradiation [35,36]. _ENREF_32Herein, the ultimate aim of this work is the design of cross-linked, on-demand dissolution chitosan hydrogel by using a one-pot, two-step reaction, and to explore the potential of diselenide-containing hydrogel for wound dressing.

## 2. Results and Discussion

### 2.1. Preparation and Characterization of Diselenide Modified Chitosan

According our previous study, diselenide-containing polymers could be successfully prepared by aminolysis of γ-selenobutyrolactone using amine, polyether amine or poly(2-ethyl-2-oxazoline-)-*co*-ethylenimine (PEtOx-EI) [37]. Chitosan, the second most natural biological macromolecule, has high content of amino group, which also could react with γ-selenobutyrolactone. As shown in Scheme 1, the selenol groups could be introduced onto the backbone of chitosan through aminolysis of γ-selenobutyrolactone with the amino group of D-glucosamine units. In order to enhance nucleophilic ability of amino group, the pH value of solution was adjusted to the isoelectric point of chitosan, leading deprotonation of amino group. In the present case, it was found that the solution with a pKa value of ~6.2 was suitable for the ring-opening reaction of γ-selenobutyrolactone with amino group of D-glucosamine units. The reaction was studied by ^1^H NMR spectroscopy, the proton signals of −CH_2_-CH_2_-CH_2_− fragment (~2.26 ppm) and –CO-CH_2_− protons (~2.50 ppm) weakened with the prolonging of reaction time (Appendix A). After 81 h, the conversion of seleobutyrolactone is up to 90%. If the reaction time was extended to 4 days, the reaction solution was too sticky to characteizer by ^1^H NMR analysis. Then the sticky mixture turned into hydrogel through its exposure to air for 1 day for the cross-linking of selenol on the side group of chitosan. The Ellman’s reagent was also used to track the reaction, which was widely applied to detect thiol and selenol (Appendix A) [35,38]. As shown in the UV-vis spectra (Appendix A), the absorption peak of compound 2 appeared around 370 nm indicating the formation of selenol. After exposure to O_2_, the peak around 370 nm did not appear, suggesting that selenol had been oxidized and diselenide had formed. The infrared absorption of chitosan was also found to obviously change (Appendix A). The absorption intensity at 2904.1 cm^−1^ decreased, and the absorption intensity at 1637.5 cm^−1^ increased, indicating the formation of amide. However, the absorption of Se–Se bond around 1000 cm^−1^ was so weak, the obvious change was not observed in the IR spectra.

To further confirm the occurrence of ring-opening reaction, the crystallinity and element content of the hydrogels were analyzed by X-ray diffraction (XRD) and X-ray photoelectron spectroscopy (XPS), respectively. XRD patterns were used to define the crystallographic pattern and atomic composition of the chitosan hydrogel by the position and intensity maxima. After purification of the hydrogel, its crystallization behavior was characterized to compare with the chitosan’s. As shown in Figure 1, chitosan had a sharp diffraction peak at 20° indicating high crystallinity [39]. After modification of γ-selenobutyrolactone, the crystallization ability of chitosan decreased with the increasing of Se content, indicating that the ordered structure of chitosan is partially destroyed. Fortunately, the reduction of intermolecular forces of chitosan contributed to good swelling property of the diselenide-containing chitosan hydrogel, which could swell facilely in pure water or phosphate buffer saline (PBS) buffer (pH = 7.4). It makes the application more convenient. The surface and near-surface atomic composition of the Se-containing chitosan hydrogel was investigated by XPS measurement. As shown in Figure 2, C, N, O and Se elements with binding energies of C 1s, N 1s, O 1s and Se 3d were observed around 285, 399, 533 and 56 eV respectively. According to the previously reported values, it could be confirmed that the chitosan was successfully modified by γ-selenobutyrolactone [40,41]. The materials were further analyzed by thermo gravimetry analyzer (TGA). As shown in Appendix A, the TGA curves of these materials showed different thermal decomposition process before and after modification of chitosan. The weight loss before 100 °C was the loss of free water in the material, and the weight loss in the range 200–360 °C was caused by the breakage of chitosan molecular chains and the loss of chemically bound water. The thermal decomposition of sugar residues is between 360 °C and 640 °C. In the decomposition process of modified chitosan, the temperature of complete decomposition was up to 800 °C. This is because chemical bonds in Se-containing chitosan hydrogels were more difficult to interrupt than pure intramolecular and intermolecular hydrogen bonds, requiring higher temperatures. All the above results suggested that the Se-containing chitosan was prepared successfully.

### 2.2. Rheological Property, Swelling Behaviour of Se-Containing CS-Se_2_ Hydrogels

The substitution degree of diselenide group on the chitosan was the essential parameter to determine the properties of Se-containing CS–Se_2_ hydrogels. Rheological studies were used to determine the chitosan hydrogel samples’ viscoelastic behaviors. Figure 3 shows the strain and frequency sweeps performed on chitosan hydrogels, with the Se-containing molar ratios (MR) of 2 % and 4 mol%, before and after swelling for 12 h in PBS buffer. A strain sweep test was first performed on both hydrogels in order to establish their linear viscoelastic domain and to compare the storage modulus values (G’) of different formulations. As shown in Figure 3a,b, G’ was higher than G’’ for both hydrogels in all cases, which means the elastic component of the material is dominant over the viscous behaviour and confirms the crosslinked structure of the samples. In the ‘as-prepared’ state, G’ increased with Se content, implying that the higher theoretical crosslink density reinforced the mechanical strength of material. Furthermore, G’ decreased after swelling for 12 h as expected for a swollen gel, while the value of G’ was still higher than G’’, demonstrating that the gel was only swollen. The difference of viscoelastic behaviours for both hydrogels was also still visible after being swollen (Figure 3a’,b’). A wide linear viscoelastic domain (wider than 0.1–10%) was observed for the two hydrogels both in the “as-prepared” state (3 wt%, Figure 3a) and after 12 h swelling in PBS buffer (Figure 3a’). The G’ of the 2 mol% Se-containing hydrogel showed small variations before and after being swollen, but those variations were limited enough to consider still in linear viscoelastic domain, implying its good manipulation and application.

Hydrogels used for wound healing should have the ability to hold fluid, which translates the capacity to remove wound exudates and maintain a moist environment for the wound sites. As swelling-controlled systems, the degree of swelling depends on crosslinking density, which is important to regulate the pore size of materials. Figure 4 shows the evolution of the swelling ratio of 2 and 4 mol% Se-containing hydrogels for different periods of time in a buffer solution. After exposure to an excess of PBS buffer at 25 °C, the swelling rate of 2 mol% Se-containing chitosan hydrogel was up to 1500% in 8 h and then decreased slightly. The possibility is that hydrogels sample was so weak after 20 h of swelling, there was quality loss in the process of weighting. The 4 mol% chitosan hydrogel swelled to an equilibrium value of 1200% in 8 h. These results showed that fluid uptake decreased with the increasing Se-contents, which confirmed the higher crosslink density. Physically, this type hydrogel exhibited a certain elasticity, and it wass soft to the touch and transparent, which is interesting for applications as wound dressings.

### 2.3. On Demand Dissolution of CS-Se_2_ Hydrogels

It is well known that diselenide bonds could respond to redox conditions. The introduction of Se–Se bond into chitosan hydrogel endows it with rapid on-demand dissolution in oxidative (H_2_O_2_) or reductive (GSH) conditions (Scheme 2). As shown in Table 1, Figure 5 and Appendix A, the 2 mol% Se-containing chitosan hydrogels samples were dissolved after 2 min and 1 min by treating with H_2_O_2_ (2 mL, 3 wt%) and GSH (2 mL, 10 mM) respectively. In addition, the dissolution time of the 4 mol% Se-containing chitosan hydrogels increased but was tolerable. No special or expensive reagents additionally were required to dissolve the hydrogel, implying its easy use in clinical treatments.

### 2.4. In Vitro Cytotoxicity Studies of the Hydrogel

Biocompatibility of a hydrogel is a crucial characteristic for its application in burn dressing. In vitro cytotoxicity of Se-containing hydrogels were evaluated by 1 days of incubation with NIH-3T3 cell. CCK8 assays were used to evaluate the cytotoxicity of the 2 mol% Se-containing chitosan hydrogel toward the NIH-3T3 cells. As shown in Figure 6, the hydrogel had a negligible effect on the cell viability (cell viabilities 95%) at a high concentration up to 10 mg mL^−1^, which confirmed that the hydrogels were potentially safe to be used as burn dressing. 

## 3. Conclusions

A novel type diselenide-linkered chitosan hydrogels was successfully synthesized using a facile ring-opening reaction of γ-selenobutyrolactone with chitosan. The crosslinking degree of chitosan hydrogels was increased with γ-selenobutyrolactone leading to a higher swelling ratio of the hydrogel. The chitosan hydrogels also show redox responsive properties and could be dissolved on-demand in a reduction or oxidation condition. For example, the hydrogel can be completely dissolved in a few minutes in H_2_O_2_ solution, a commonly oxidant using as antiseptic or disinfectant in clinic. In addition, by adding an excess of GSH solution, a peptide widely existing in animals, plants and microbials, the hydrogel can be also completely dissolved. This is a very important feature considering that burn wound dressings need to be changed regularly. The dressing is soft to adapt perfectly to the wound shape and keep it moist at all times. All these properties combining with low toxicity make this material very interesting for burn wound dressing applications.

## 4. Materials and Methods

### 4.1. Materials 

Chitosan (CS, degree of deacetylation: ~95%, viscosity: 100–200 mpas) was purchased from Shanghai Macklin Biochemical Co., Ltd., Shanghai, China. Hydrogen peroxide (H_2_O_2_, 30 wt%) was purchased from Yonghua Chemical Technology (Jiangsu) Co., Ltd. (Suzhou, China) Acetic acid (99.5%) was purchased from Shanghai Chemical Reagent Co., Ltd., Shanghai, China, and used as received. Glutathione (Reduced, GSH, 98%) was purchased from Shanghai Aladdin biochemical technology Co., Ltd., Shanghai, China. γ-Selenobutyrolactone [42] was synthesized according to the protocol described in the reference. NIH3T3 were gifted by Prof. Haibin Shi’s group and were cultured in Dulbecco’s modified Eagle medium (DMEM; Hyclone, Logan, UT, USA) containing 10% fetal bovine serum (FBS; Hyclone).

### 4.2. Preparation of Se-Containing Chitosan Hydrogels

1.84 g chitosan and 48.50 mL acetic acid (2.10 wt%) were added in a 100 mL flask. The NaOH aqueous solution (4 wt%) was dropped slowly to adjust the pH value to 6.20. Then γ-selenobutyrolactone (32.40 mg) was injected into deoxidized chitosan solution (60.34 g) by microinjector. The reaction mixture was stirred for 4 days at 50 °C under argon. Se-containing chitosan (CS–Se_2_) hydrogel was obtained after exposing it in air for 1 day.

### 4.3. Structural Characterization of Se-Containing Chitosan Hydrogels 

The Se-containing chitosan hydrogels were dialyzed (MWCO 1000 Da) in deionized water for 24 h at 25 °C in order to remove the inorganic salt, and freeze-dried to constant weight. X-ray photoelectron spectroscopy (XPS) analysis of the samples was carried out using ESCALAB 250 XI (Thermo Fisher Scientific, Al KR source, Waltham, MA, USA). The general spectra in the range 0–1100 eV and narrow spectra at high resolution for all elements were recorded. The wide angle X-ray diffraction (WAXD) patterns of the hydrogels were obtained using an PANalytical X’Pert-Pro MPD X-ray diffractometer (Egham, Surrey, UK). Test parameters: copper target CuKα (λ = 0.15406), power 1600W (40 kW, 40 mA), 2θ = 10°–65 °, scan speed = 4° min^−1^, step size = 0.02°.

### 4.4. Rheology Characterization of Se-Containing Chitosan Hydrogels

The chitosan hydrogel (1.5 g) was swelled in PBS buffer (pH = 7.4) for 12 h to reach the equilibrium. All rheological measurements were performed at 25 °C. Samples were analyzed in parallel plate geometry (rotor size: 20 mm). The gap between the sample and the rotor is 0.4 mm to insure a good contact. The dynamic strain sweep was performed from 0.001% to 10% strain with frequency at 1 Hz. The dynamic frequency sweep test was conducted from 0.1 Hz to 100 Hz with strain at 1.0%. All measurements were made in triplicate for each hydrogel using separate samples and the mean SEM of the values was reported.

### 4.5. Swelling Behavior of Se-Containing Chitosan Hydrogels

Chitosan hydrogels with difference Se-containing were weighed and placed in sealed 200-mesh nylon mesh. The hydrogel was then immersed in PBS buffer (pH = 7.4) at 25 °C. The degree of swelling of the chitosan hydrogels was calculated by weighing the hydrogel after removing the entire PBS at predetermined time intervals. The swelling ratio (SR) of the hydrogel was determined using the following equation:$$\%\mathrm{SR}=\frac{\left(W_{t}-W_{0}\right)}{W_{0}}\times 100$$
where *W_t_* and *W*_0_ are the weights of the swollen gel and the initial sample, respectively. All the experiments were carried out three times using separate samples, and the average values were reported.

### 4.6. The Redox Responsive Behavior of Se-Containing Chitosan Hydrogels

The Se-containing chitosan hydrogels were immersed in PBS buffer (pH = 7.4) for 12 h at 25 °C. 2 g of them were weighed into clear vials and dissolved by 2 mL H_2_O_2_ (3 wt%) and 2 mL GSH (10 mM) at 25 °C, respectively. The dissolution time was recorded when flow phenomena was observed.

### 4.7. In Vitro Cytotoxicity Assay

Biocompatibility is an important property for wound dressings. The cytotoxicity of both hydrogels was studied with NIH3T3 cells, which were cultured in Dulbecco’s modified Eagle’s medium (DMEM) and supplemented with 10% fetal bovine serum (FBS) and 1% penicillin/streptomycin. After plating, the cells were incubated in a 5% CO_2_ incubator at 37 °C, with culture medium changed every 2 days.

After pre-sterilized under UV light for 1 h, the hydrogel samples were divided into five groups. The extracts (2, 4, 6, 8 and 10 mg mL^−1^) of the hydrogel samples were made by dipping the samples with DMEM medium for 72 h. The concentration of chitosan used was the same as hydrogel samples. Furthermore, blank group (culture medium) and control group (culture medium and cells, without chitosan or Se-containing hydrogels) were set to compare with the experimental group. The NIH-3T3 cells were seeded on 96-well plates at a density of 8 × 10^3^ well^−1^ and incubated at 37 °C in a humidified atmosphere of 5% CO_2_. Culture medium and extracts were removed after 24 h. Then 10 μL CCK-8 solution and 90 μL culture medium were added to each well, and the plates were further incubated for 2 h at 37 °C. Sample’s absorbance was measured at 450 nm using a microplate reader and the survival rate was calculated.

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
