# Peer review of "An On-Demand Dissoluble Chitosan Hydrogel Containing Dynamic Diselenide Bond"

_gels, 2021, doi:10.3390/gels7010021_

Round 1
Reviewer 1 Report
The worst flaw of this work lies in the fact that no scientific evidence is ever given of the formation of a C-Se bond and especially Se-Se. The measurements made with different techniques say nothing about the reaction
Author Response
Dear Editor and Reviewer,
Thank you very much for giving us a chance to revise this paper. We also thank the reviewers for their critical and professional comments for improving the quality of the manuscript. We have made specific efforts according to the comments/suggestions of the reviewers, and the manuscript has been revised adequately based on the supplemental results and a careful literature survey. The questions put forward by the reviewers have been answered individually below.
Q1: The worst flaw of this work lies in the fact that no scientific evidence is ever given of the formation of a C-Se bond and especially Se-Se. The measurements made with different techniques say nothing about the reaction.
A1: Thanks very much for your professional comment. Aminolysis of selenolactone has been reported in our previous works. The reaction was carefully investigated using NMR and MS. The amino group of chitosan is confident of success to open the selenolactone. As show in the 1H NMR spectrum, the content of selenolactone was decrease with time indicating the ring opening reaction of selenolactone. The Ellman’s reagent was also used to detect the formation of selenol (Ellman, G. L., A colorimetric method for determining low concentrations of mercaptans. Arch. Biochem. Biophys. 1958, 74, 443-450). As shown in the UV-vis spectra, the absorption peak of compound 2 appeared around 370 nm indicating the formation of selenol. After oxidation of O2, the peak around 370 nm did not appear indicating the diselenide was formed. Other circumstantial evidence, such as FTIR spectra, XRD, XPS, redox-responsive property rheology characterization of the hydrogel, are supported the formation of diselenide.

Reviewer 2 Report
The overall article looks good and very relevant to the area of research. It requires major revision. Please find the attachment for further comments.

Author Response
Dear Editor and Reviewer,
Thank you very much for giving us a chance to revise this paper. We also thank the reviewers for their critical and professional comments for improving the quality of the manuscript. We have made specific efforts according to the comments/suggestions of the reviewers, and the manuscript has been revised adequately based on the supplemental results and a careful literature survey. The questions put forward by the reviewers have been answered individually below.
Finally, due diligence should be applied. Please write full sentences, cut out repetitions, run a spell-checker, and have it revised. The article need major corrections and need to have complete alignments.
Q1. What is the minimal concentration of Se that could be administered other than 2% or 4%?
A1:Thanks very much for your professional comment. The concentration of Se could be adjust by adding selenolactone. We also prepared 1 mol% and 6 mol% Se-containing CS-Se2 hydrogels. However, the 1 mol% Se-containing CS-Se2 hydrogel was so weak for application, and swelling ratio of 6 mol% Se-containing CS-Se2 hydrogel is too low to use as dressing.
Q2. The article contains various characterization to confirm the presence of diselenide-containing polymers (chitosan). To further increase the confirmation, FT-IR characterization could be taken before and after the formation of diselenide with chitosan to understand the chemical bond shift.
A2: Thanks very much for your professional comment. The FT-IR spectra was supplied in the supporting information. The intensity of IR absorption of at 2904.1 cm-1 decreased, and the intensity of IR absorption at 1637.5 cm-1 increased indicating the formation of amide. However, the absorption of Se-Se bond around 1000 cm-1 was so weak; the obvious change was not observed in the IR spectra.
Q3. With the XRD characterization, please explain in more details regarding the crystallinity peaks of standard chitosan and chitosan with Se2.
A3: Thanks very much for your professional comment. XRD patterns were used to define the crystallographic pattern and atomic composition of the sample material by the position and intensity maxima. As shown in Figure 1, chitosan has a sharp diffraction peak at 20o indicating the characteristic peak of chitosan flecks. After modification of γ-selenobutyrolactone, some chemical crosslinks are produced between the chitosan chains resulting in the decrease of the regularity of chitosan. Moreover, the crystallization ability of chitosan decreases with the increasing of Se content.
Q4: The articles says the ordered structure of chitosan is partially destroyed with the increasing of Se content. Does the chitosan role be maintained with the Se content for the purpose of application?
A4: Thanks very much for your professional comment. The chain of chitosan is regular, and there are many intramolecular and intermolecular hydrogen bonds. It makes the chitosan hard to dissolve pure water or PBS buffer (pH = 7.4). We used acetic acid solution (~2 wt%) to dissolve the chitosan due to the acid environment could destroy the intramolecular and intermolecular hydrogen bonds. However, this condition is not a good idea for application. Fortunately, the intermolecular forces of chitosan could be reduced by the modification of diselenide, and the diselenide-containing chitosan hydrogel could swell facilely in pure water or PBS buffer (pH = 7.4). It makes the application more convenient.
Q5. The In vitro cytotoxicity studies of the hydrogel is demonstrated with 2 mol% Se-containing chitosan hydrogel. In fig.6, there are no controls. Evaluate the cytotoxicity text with just chitosan and existing controls to show more detailed efficacy on the cytotoxicity studies.
A5: Thanks very much for your professional comment. The in vitro cytotoxicity experiment of the hydrogel was repeated. The controls were added to show biocompatibility of the hydrogel. All the modification in the main text are highlight.
Q6: Figure legends could be maintained same for all the figures
A6: All of the Figure legends are adjusted. All the modification in the main text are highlight.
Q7: Need to format all the sub title and must align with the content
A7: Thanks very much for your professional comment. All the sub title are checked.
Q8: Abbreviations of all the content in the article should be provided.
A8: Thanks very much for your professional comment. All the content in the article are check. Some abbreviations are added.

Round 2
Reviewer 1 Report
For me now it is acceptable
Reviewer 2 Report
The article looks good with the provided additional information.